# The greenbeard gene *tgrB1* regulates altruism and cheating in *Dictyostelium discoideum*

Mariko Katoh-Kurasawa[1], Peter Lehmann[1,2] & Gad Shaulsky [1] ✉

Greenbeard genetic elements encode rare perceptible signals, signal recognition ability, and altruism towards others that display the same signal. Putative greenbeards have been described in various organisms but direct evidence for all the properties in one system is scarce. The *tgrB1-tgrC1* allorecognition system of *Dictyostelium discoideum* encodes two polymorphic membrane proteins which protect cells from chimerism-associated perils. During development, TgrC1 functions as a ligand-signal and TgrB1 as its receptor, but evidence for altruism has been indirect. Here, we show that mixing wild-type and activated *tgrB1* cells increases wild-type spore production and relegates the mutants to the altruistic stalk, whereas mixing wild-type and *tgrB1*-null cells increases mutant spore production and wild-type stalk production. The *tgrB1*-null cells cheat only on partners that carry the same *tgrC1*-allotype. Therefore, TgrB1 activation confers altruism whereas TgrB1 inactivation causes allotype-specific cheating, supporting the greenbeard concept and providing insight into the relationship between allorecognition, altruism, and exploitation.

Greenbeards were originally proposed as hypothetical selfish genetic elements to illustrate how cooperation might be maintained despite the apparent cost of altruism[1,2]. Greenbeards have been considered unlikely because of their complexity, but empirical studies have shown the existence of various putative greenbeard types in the real world[3]. Nevertheless, many of these examples fall short of fulfilling all the greenbeard properties and discrepancies between theory and experiments have raised the need for additional empirical evidence[4].

Two putative greenbeard examples have been described in *D. discoideum*. These soil amoebae propagate as unicellular organisms when food is abundant. Upon starvation, propagation stops, and the cells aggregate into a cooperative multicellular structure in which 80% of the cells become viable spores and 20% die while forming a cellular stalk[5]. This is a form of altruism because the stalk cells sacrifice themselves while helping in spore dispersal. The developmental process is risky because *D. discoideum* form chimeras that expose cells to exploitation by cheaters – strains that generate more spores than their

fair share[6–8]. The developmental cell adhesion gene *csaA* was described as a greenbeard because its bearers cooperate with one another, whereas the absence of the gene leads to cheating[9,10]. This system is not a perfect greenbeard, however, because the *csaA* gene is not polymorphic so it does not exhibit an unusual signal that could distinguish kin from non-kin[4]. The *tgrB1-tgrC1* allorecognition system is also a greenbeard[11]. *tgrB1* and *tgrC1* are linked polymorphic genes that encode single-pass transmembrane proteins[12]. There is strong evidence for their role in allorecognition in lab experiments[12,13] and in nature[11], but the evidence for altruism is indirect[4,11]. Laboratory experiments have shown that *tgrB1* and *tgrC1* are among the most polymorphic genes in the *D. discoideum* genome[12,14]. Together, they are necessary and sufficient for allorecognition[12,13], and they function as a ligand-receptor pair in multicellular development[15]. Many of the laboratory experiments have used engineered strains that only differed in their *tgrB1-C1* loci, but *tgrB1-C1* sequence polymorphism correlates well with the segregation of wild strains, suggesting that the

[1]Department of Molecular and Human Genetics, Baylor College of Medicine, Houston, TX 77030, USA. [2]Graduate program in Genetics and Genomics, Baylor College of Medicine, Houston, TX 77030, USA. ✉e-mail: gadi@bcm.edu

allorecognition observed in the laboratory is relevant in nature as well[11,12,16].

The role of *tgrB1-C1* in sociality has been studied in several contexts. One study showed that allorecognition can protect cooperators against cheaters caused by mutations outside the *tgrB1-C1* locus[17]. Another showed that it protects cells from adverse interactions during slug migration[11]. Others have shown that the *tgrB1-C1* locus defines kinship among natural isolates of *D. discoideum*[11,12,16]. The genes also have developmental roles, because they are essential for tissue integration and spore and stalk production[12,15,18-20]. Nevertheless, there has been no evidence for their direct involvement in altruism or cheating to our knowledge.

A matching pair of *tgrB1* and *tgrC1* encodes proteins that bind each other and mediate development and allorecognition[11,15]. Strains that carry two sets of different *tgrB1-C1* allotypes develop well and cooperate with cells of both allotypes, suggesting that allorecognition is inclusive rather than exclusive[13]. In contrast, a mismatching pair of *tgrB1* and *tgrC1* is incompatible with normal development, and strains that carry such pairs behave like the respective null strains. A screen for genetic suppressors of such a mismatch revealed dominant mutations that activate the *tgrB1* gene product[21]. The mutations suppressed the original *tgrB1-C1* mismatch as well as mutations that inactivate *tgrC1* (*tgrC1⁻*) or both genes (*tgrB1⁻tgrC1⁻*)[15,21]. Therefore, the mutant TgrB1 protein can exert its receptor activity in the absence of its ligand. The *tgrB1* gene is highly polymorphic in natural populations and the proteins it encodes vary up to 13% in their amino acid sequences[11,12]. The polymorphism is not distributed evenly throughout the gene. The region that encodes the intracellular domain is nearly invariable, and the regions that encode the immunoglobulin folds of the extracellular domain are much less variable than the other extracellular regions[12]. None of the activating mutations found in the screen matched any naturally occurring SNPs in the *tgrB1* coding sequences. Most of them were in the extracellular domain, and one was in the highly conserved intracellular domain. We used two of these mutations to test the hypothesis that *tgrB1* activation might cause altruism, the L846F mutation that modified the intracellular domain next to S845, which is a phosphorylation site of unknown function[15], and the G275D mutation that resides in the 5′ end of the region that encodes the first immunoglobulin fold[12].

Here, we show that activation of the TgrB1 receptor confers altruism in that cells that carry the activated *tgrB1* allele produce more of the prestalk and stalk cells when mixed with wild-type cells. The wild-type cells also produce more spores in the chimeric structures than they do in pure populations. We also show that inactivation of the *tgrB1* gene causes cheating and that cheating is allotype-specific. These findings support the conclusion that *tgrB1* and *tgrC1* encode a green beard system that fulfills all the predicted criteria.

## Results

### Activation of *tgrB1* confers altruism

We co-developed an RFP-marked (Red Fluorescent Protein) strain that carries the activated allele *tgrB1^L846F^* with a wild-type counterpart, GFP-marked (Green Fluorescent Protein) AX4, which has an intact pair of *tgrB1-C1* alleles. The activated *tgrB1* strain became enriched in the anterior (tip) and posterior (rearguard) regions at the finger stage (Fig. 1a) and in the tip during culmination (Fig. 1b). These regions normally consist of prestalk cells[22,23]. To further explore the relationship between the activated *tgrB1* strain and its AX4 mixing partner, we expressed the activate allele *tgrB1^L846F^* in cells constitutively labeled with *[act15]*:CFP (Cyan Fluorescent Protein) and mixed them with AX4 cells tagged with the prestalk reporter *[ecmA]*:YFP (Yellow Fluorescent Protein) and the prespore reporter *[cotB]*:RFP. In this mix, the *tgrB1^L846F^* cells were also enriched in the prestalk regions and intermixed with the YFP-tagged AX4 prestalk cells, whereas the central prespore region was mainly occupied by the RFP-tagged AX4 prespore cells (Fig. 1c). A control mix of dual-tagged AX4 cells with constitutive CFP-labeled AX4 cells showed that the CFP-positive cells were represented throughout the developmental structure, overlapping with both the prespore and prestalk markers of the mixing partner (Fig. 1d). There were no overt differences in the prespore-prestalk ratios and positions of the labeled AX4 cells in the two mixes (Fig. 1c, d). These results suggest that the activated *tgrB1* strain preferentially assumed the prestalk fate when mixed with AX4.

To test that possibility further, we developed pure and mixed strains, differentially labeled with GFP and RFP, and counted the spore production of each strain. Figure 1e shows that AX4 increased its spore production in the mix about 24% compared to its spore production in a pure population. This finding suggests that the activated *tgrB1* strain altruistically increased the spore production of its mixing partner. We repeated the spore production test with another activated allele, *tgrB1^G275D^*, and found again that AX4 produced about 33% more spores in the mix than it did in the pure population (Supplementary Fig. 1). In both cases, we did not observe significant changes in the sporulation of the activated *tgrB1* strain. The experiments shown in Fig. 1 suggest that *tgrB1* activation confers altruistic behavior, which is manifested as increased contribution of the activated *tgrB1* strain to the prestalk region and increased sporulation of the wild-type counterpart in mixed populations. These observations support the hypothesis that *tgrB1* is a greenbeard element whose activation is sufficient to confer altruistic behavior.

### Inactivation of *tgrB1* confers cheating

If activation of *tgrB1* confers altruism, its inactivation might cause cheating. To test that possibility, we co-developed differentially-labeled AX4 and *tgrB1⁻* cells, both of which carry the same *tgrC1* allele. The *tgrB1⁻* cells became enriched in the central and posterior regions at the finger stage (Fig. 2a) and in the spore-bearing sorus during culmination, as well as in the basal disk (Fig. 2b). The *tgrB1⁻* cells were largely excluded from the anterior finger region (Fig. 2a) and from the culminant tip and stalk (Fig. 2b). The wild-type counterpart was the main occupant of the finger anterior (Fig. 2a), as well as the tip and stalk during culmination (Fig. 2b). These results suggest that the *tgrB1⁻* strain preferentially assumed the prespore fate whereas AX4 preferentially assumed the prestalk fate in mixed development. We then tested the effect of the *tgrB1⁻* strain on the development of the AX4 victim by mixing *tgrB1⁻* cells constitutively labeled with *[act15]*:CFP with AX4 cells tagged with the prestalk reporter *[ecmA]*:YFP and the prespore reporter *[cotB]*:RFP. In this mix, the *tgrB1⁻* cells were found mainly in the posterior region, intermixed with the RFP-tagged AX4 prespore cells, whereas the anterior prestalk region was mainly occupied by the YFP-tagged AX4 prestalk cells (Fig. 2c). Despite the altered distribution of the two strains, there were no obvious differences in the prespore-prestalk ratios and positions of the tagged AX4 cells in the mix. The *[cotB]*:RFP prespore cells occupied the posterior finger region and the *[ecmA]*:YFP prestalk cells occupied the anterior finger region (Fig. 2c), similar to their positions in the control mix (Fig. 1d).

To further test the consequences of the interaction between the wild-type and mutant cells, we developed pure and mixed strains and counted their spore production. Figure 2d shows that the *tgrB1⁻* strain formed fewer than $2 \times 10^5$ spores (representing less than 3% sporulation efficiency) when developed in a pure population, but its sporulation increased more than sevenfold, to $1.4 \times 10^6$, when mixed with AX4. In addition, the presence of the *tgrB1⁻* strain caused a 24% reduction in the AX4 sporulation efficiency. We also tested the consequences of mixing the *tgrB1⁻* strain with the activated *tgrB1^L846F^* strain (Supplementary Fig. 2). In this mixing experiment, the *tgrB1⁻* strain formed $1.6 \times 10^5$ spores when developed in a pure population, but its sporulation efficiency increased more than eightfold, to $1.4 \times 10^6$, when mixed with the *tgrB1^L846F^* strain. While this effect was very similar to the mixing with AX4, the effect on the activated *tgrB1^L846F^* strain was

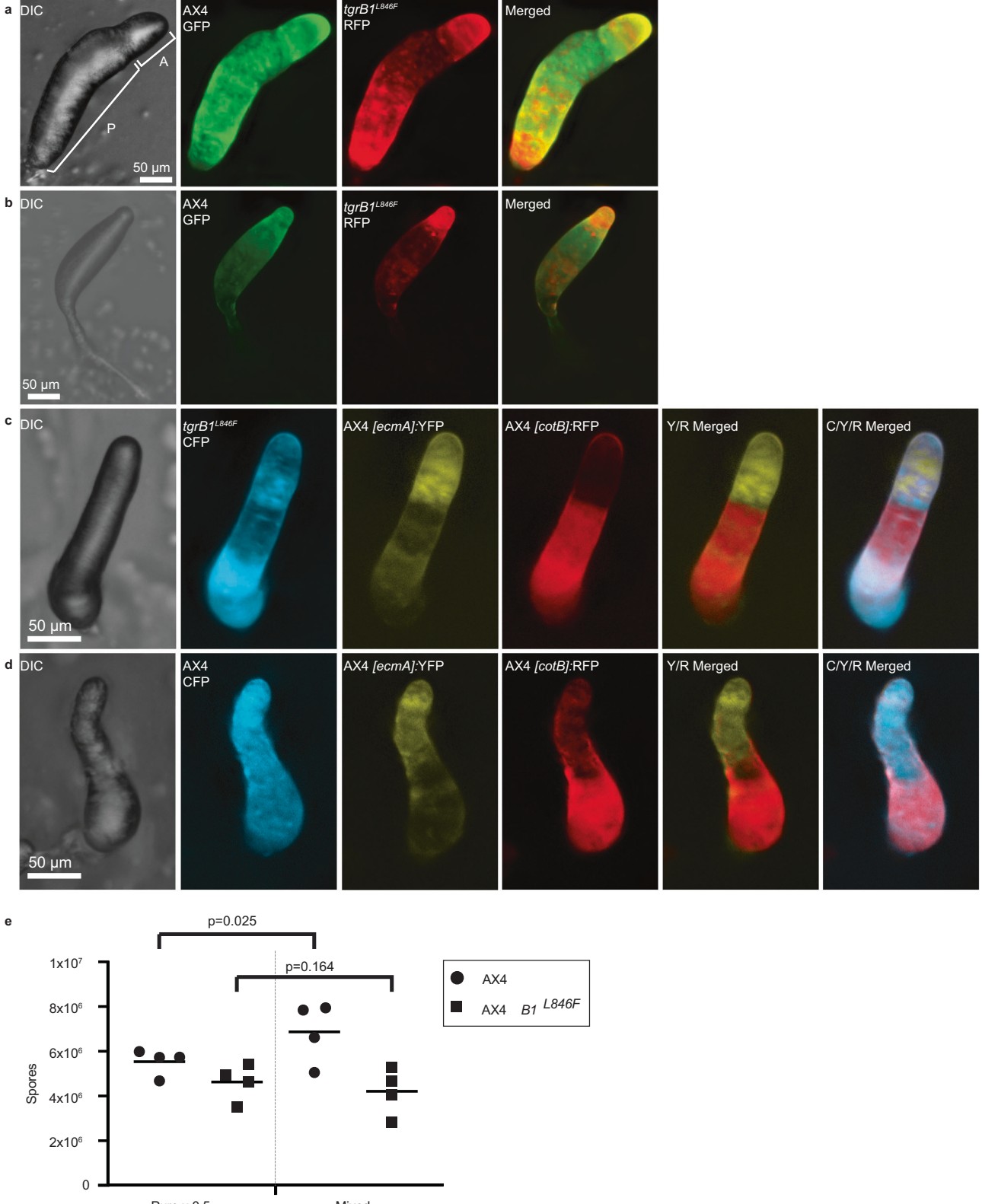

greater. The activated *tgrB1^{L846F}* spore production was reduced more than 1.5 fold, from $4.9 \times 10^6$ spores in the pure population to $3.1 \times 10^6$ spores in the mix with the *tgrB1^-* strain (Supplementary Fig. 2).

These experiments suggest that the absence of *tgrB1* results in a partial cheating behavior, which is manifested as increased propensity of the *tgrB1^-* cells to occupy the prespore region and to produce

spores. The wild-type counterpart incurred a cost that was seen as a disproportional propensity to contribute to the stalk as well as decreased spore production. This cost was even more pronounced when the counterpart expressed the activated *tgrB1* allele. This partial cheating behavior is consistent with the proposed role of *tgrB1* as a greenbeard element, which is necessary for altruism.

**Fig. 1 | Activated *tgrB1* confers altruistic behavior.** We used two strains that express constitutive fluorescent markers, the wild-type AX4-GFP (green) and the activated *tgrB1* mutant AX4 *tgrB1^L846F^*-RFP (red). We grew the cells separately, mixed equal proportions, and co-developed them. We imaged the structures at the finger (**a**) and culminant stage (**b**) with DIC and with green and red fluorescence, and generated a merged image of the red and green channels as indicated. The brackets in panel **a** show the anterior region (A) that contains mainly prestalk cells and the posterior region (P) that contains mainly prespore cells. **c** We also co-developed constitutively labeled mutant *tgrB1^L846F^*-CFP cells (cyan) with wild-type AX4 cells carrying the prestalk reporter *[ecmA]*:YFP (yellow) and the prespore reporter *[cotB]*:RFP (red). We imaged the structures with DIC and with cyan, yellow, and red fluorescence, and generated merged images of the yellow and red (Y/R) as well as all three channels (C/Y/R) as indicated. **d** As a control, we co-developed constitutively labeled wild-type AX4-CFP cells (cyan) with wild-type AX4 cells carrying the same prestalk and prespore reporters and imaged them as above. **e** We grew wild-type AX4-GFP and mutant *tgrB1^L846F^*-RFP cells separately, developed 7×10^6 cells either in pure populations or mixed at equal proportions as indicated, and counted spores. The spore counts are shown as four independent replicates (symbols) and their averages (horizontal lines). The pure population counts were multiplied by 0.5 to scale them with the mixed population. Brackets and p-values (T-test, one-sided, *n* = 4) compare the spore counts of each strain in the two conditions. Camera settings are included in Supplementary Data 1. Source data are provided as a Source Data file.

## Developmental consequences of cheating by *tgrB1⁻*

To further explore the developmental relationship between the *tgrB1⁻* cheater and the AX4 victim, we mutated *tgrB1* in a strain that carries the prestalk reporter *[ecmA]*:GFP and the prespore reporter *[cotB]*:RFP. Pure *tgrB1⁻* cells do not express *ecmA* during development and they express very low amounts of *cotB* at late developmental stages but not at 16 hours, which is the wild-type finger stage (Supplementary Fig. 3). Mixing the double-tagged *tgrB1⁻* strain with unlabeled AX4 resulted in expression of both prespore and prestalk markers in the mutant at the finger stage. The *tgrB1⁻ [ecmA]*:GFP prestalk cells were largely excluded from the anterior prestalk region. Instead, they were found mainly in the area that normally spans the prestalk-prespore border (Fig. 3a). The *tgrB1⁻ [cotB]*:RFP prespore cells exhibited punctate staining, suggesting that only a few *tgrB1⁻* cells expressed detectable levels of the prespore marker (Fig. 3a). These cells were mainly localized in the posterior part of the finger, which is the prespore region. We also observed a narrow region of overlapping staining between the prespore and prestalk marked cells (Fig. 3a), which is not normally found in the wild type (Fig. 3b). This overlapping staining is not likely due to transdifferentiation. There is a very small number of co-labeled *[cotB]*:RFP and *[ecmA]*:GFP cells in the double-tagged wild type, and we did not observe a change in that number in the double-tagged *tgrB1⁻* strain.

As a control, we compared the expression of the reporters in the wild type and the mutant in pure populations. In the AX4 wild type, the green prestalk cells occupy the anterior region and the red prespore cells occupy the posterior region of the finger structure (Supplementary Fig. 4a). Most of the *tgrB1⁻* cells are found in loose aggregates at the same time (16 hours of development) and most of them do not express detectable levels of *[cotB]*:RFP and *[ecmA]*:GFP, with the exception of a few cells (arrows, Supplementary Fig. 4b). In very rare cases and in later times (20-24 hours of development), the *tgrB1⁻* cells form tipped aggregates and fingers that exhibit significant *[ecmA]*:GFP expression in the anterior and weak *[cotB]*:RFP expression in the posterior region (Supplementary Fig. 4c).

The results shown in Fig. 3 suggest that the cheating behavior of *tgrB1⁻* is correlated with its lack of contribution to the altruistic prestalk tissue, even though it is capable of differentiating into prestalk cells. They also suggest that *tgrB1⁻* cells receive signals from the wild type that induce them to express prespore and prestalk markers, albeit at levels lower than the wild type. Despite the altered spatial distribution of the mutant in the mixed structures, we found no evidence for enhanced transdifferentiation among the developing *tgrB1⁻* cells.

## Cheating by *tgrB1⁻* is allotype-specific

The greenbeard effect predicts that social behaviors should be allotype specific. We tested that prediction using strains with different allotypes. AX4 *B1^QS31^C1^QS31^* is an AX4 strain in which the resident *tgrB1-tgrC1* locus was deleted (AX4 *B1^Δ^C1^Δ^*) and then replaced with a *tgrB1-tgrC1* locus from the incompatible strain QS31[13]. We also used AX4 *B1^AX4^C1^AX4^*, in which the replacement locus was from AX4. This strain underwent the same genetic manipulations as AX4 *B1^QS31^C1^QS31^*, so the two strains differ only in their allotypes[13]. As mixing partners, we used the respective *tgrB1*-deletion strains. AX4 *B1^Δ^C1^AX4^* is a *tgrB1*-null strain in which *tgrC1* is of the AX4 allotype, and AX4 *B1^Δ^C1^QS31^* is a *tgrB1*-null strain in which *tgrC1* is of the QS31 allotype. We co-developed each of the *tgrB1*-null strains with each of the two double-gene replacement strains and compared the sporulation efficiencies to the respective pure populations. Figure 4a shows that the AX4-type *tgrB1*-null strain AX4 *B1^Δ^C1^AX4^* partially cheated on the compatible wild-type strain AX4 *B1^AX4^C1^AX4^*. This finding is similar to the one shown in Fig. 2, suggesting that the gene replacement did not alter the social interactions between the strains. Figure 4b shows that the QS31-type *tgrB1*-null strain AX4 *B1^Δ^C1^QS31^* did not cheat on the incompatible wild-type strain AX4 *B1^AX4^C1^AX4^*, suggesting that an incompatible *tgrC1* allele is inconsistent with cheating. Figure 4c shows that the AX4-type *tgrB1*-null strain AX4 *B1^Δ^C1^AX4^* did not cheat on the incompatible wild-type strain AX4 *B1^QS31^C1^QS31^*, further supporting the hypothesis that an incompatible *tgrC1* allele is inconsistent with cheating. Figure 4d shows that the QS31-type *tgrB1*-null strain AX4 *B1^Δ^C1^QS31^* partially cheated on the compatible wild-type strain AX4 *B1^QS31^C1^QS31^*, further supporting the allotype-specificity hypothesis and suggesting that cheating by *tgrB1*-null cells is consistent with the QS31 allotype and not peculiar to the AX4 allotype. The control in Fig. 4e shows that the double-null strain AX4 *B1^Δ^C1^Δ^* did not cheat on the wild-type strain AX4 *B1^AX4^C1^AX4^*, suggesting that the absence of *tgrC1* is similar to the presence of an incompatible *tgrC1* in terms of cheating specificity. The experiments shown in Fig. 4 suggest that a compatible *tgrC1* allele is required for social interactions in which *tgrC1* encodes the allorecognition signal that directs the social behavior toward compatible kin, and *tgrB1* encodes the allorecognition perception. This finding is also consistent with the developmental roles of *tgrB1* as a receptor and *tgrC1* as its ligand[15].

## Discussion

Previous studies have shown that the two linked genes, *tgrB1* and *tgrC1*, encode polymorphic membrane proteins that mediate allorecognition[12,13], and developmental studies showed that TgrC1 is a ligand and TgrB1 is its receptor[15,19]. The data shown here suggest that the *tgrB1-tgrC1* locus fulfils all the criteria of a greenbeard system[3,4]. Our results expand on the previous findings by showing that *tgrC1* encodes the perceptible greenbeard signal and *tgrB1* encodes a receptor that confers altruism toward kin upon signal recognition. The previous studies provided strong support for the greenbeard hypothesis, but they linked the *tgrB1-tgrC1* locus to somewhat general aspects of cooperation[11,17]. The findings that *tgrB1* activation causes altruism and *tgrB1* inactivation causes cheating against kin provide the missing direct evidence.

The altruistic action caused by activated *tgrB1* was observed in two ways. First, the sporulation efficiency of the wild-type cells increased in the mixed population. Although the increase might seem modest at the 1:1 mixing ratio, it is in fact quite significant because in subsequent generations the wild-type spore proportion is expected to increase exponentially as its frequency in the population increases. Second, the activated *tgrB1* cells were enriched in the prestalk region. It is somewhat surprising that we did not observe

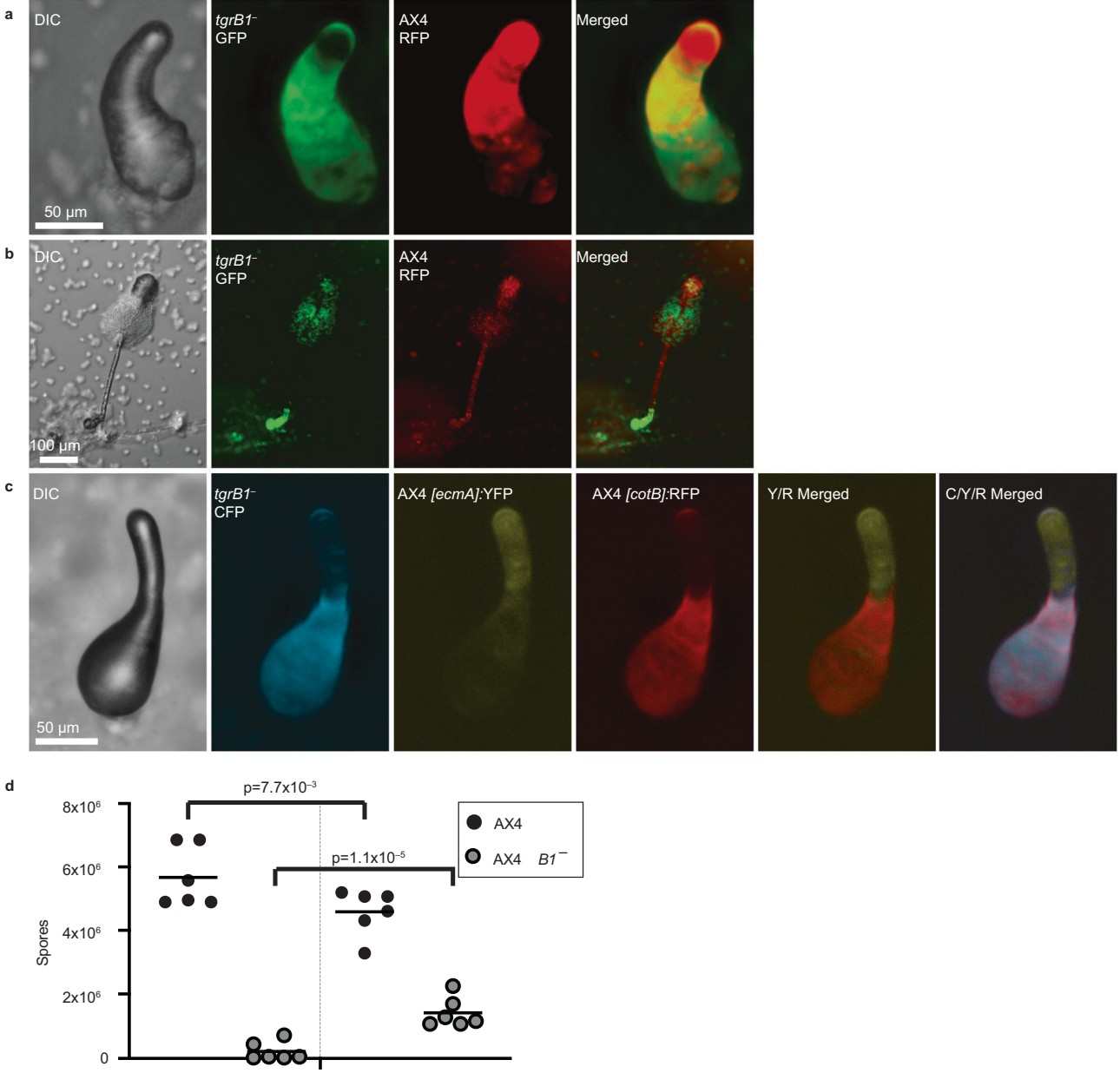

**Fig. 2 | Inactivation of *tgrB1* confers cheating behavior.** We used two strains that express constitutive fluorescent markers, the wild-type AX4-RFP (red) and the inactivated *tgrB1* mutant AX4 *tgrB1⁻*-GFP (green). We grew the cells separately, mixed equal proportions, and co-developed them. We imaged the structures at the finger stage (**a**) and fruiting body stage (**b**) with DIC and with green and red fluorescence, and generated a merged image of the red and green channels, as indicated. The green staining at the edge of the anterior region is due to reflection and is not associated with cells. **c** We also mixed constitutively labeled mutant *tgrB1⁻*-CFP cells (cyan) with wild-type AX4 cells carrying the prestalk reporter *[ecmA]*:YFP (yellow) and the prespore reporter *[cotB]*:RFP (red). We imaged the structures with DIC and with cyan, yellow, and red fluorescence, and generated

merged images of the yellow and red (Y/R) as well as all three channels (C/Y/R), as indicated. **d** We grew the cells separately, developed $7 \times 10^6$ cells either in pure populations or mixed at equal proportions as indicated, and counted spores. The spore counts are shown as six independent replicates (symbols) and their averages (horizontal lines). In this case, we used 4 different alleles of AX4 *tgrB1⁻*-GFP mixed with AX4-RFP and one of AX4 *tgrB1⁻*-RFP mixed with AX4-GFP in 6 experiments. The pure population counts were multiplied by 0.5 to scale them with the mixed population. Brackets and p-values (T-test, one sided, $n = 6$) compare the spore counts of each strain in the two conditions. Camera settings are included in Supplementary Data 1. Source data are provided as a Source Data file.

significantly reduced sporulation, but there are a few possible explanations, including differentiation of 'differentiation-null-cells' that would otherwise not contribute to the spores or stalks[24], recruitment of 'loners' that would otherwise not aggregate[25,26], or social effects that do not alter cell-type proportioning[27]. These observations were made with two different activated *tgrB1* alleles, suggesting that they reflect a natural property of *tgrB1* rather than an atypical new function.

The exploitation caused by *tgrB1* inactivation was also manifested in two ways. First, the *tgrB1⁻* cells did not contribute significantly to the prestalk and stalk tissues in mixes with the wild type, while the wild type became the major contributor to these altruistic tissues. Second, the *tgrB1⁻* cells partially cheated by making more spores in the mix than they did in the pure population and by reducing the spore production of their wild-type partners. These behaviors satisfy the general definition of cheating, namely, benefiting from a social trait without

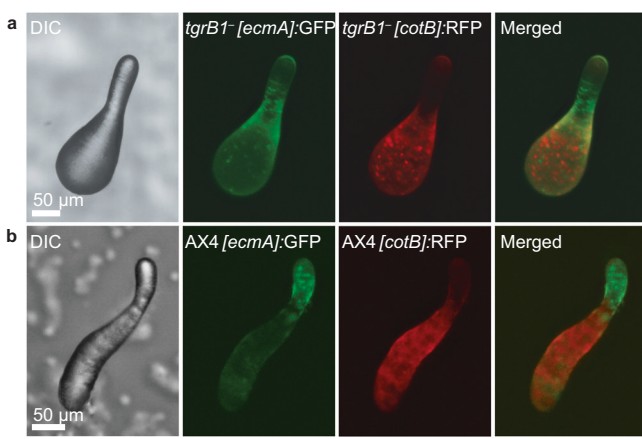

**Fig. 3 | Cell-type development of the *tgrB1⁻* cheater and its victim.** We grew strains separately, mixed them at equal proportions, co-developed them, and imaged at the finger stage. **a** We mixed unlabeled wild-type AX4 cells with mutant *tgrB1⁻* cells carrying the prestalk reporter *[ecmA]*:GFP and the prespore reporter *[cotB]*:RFP. **b** As a control, we mixed equal proportions of unlabeled AX4 cells with AX4 cells carrying the prestalk reporter *[ecmA]*:GFP and the prespore reporter *[cotB]*:RFP. We imaged the structures with DIC and with green and red fluorescence, and generated a merged image of the red and green channels, as indicated. Camera settings are included in Supplementary Data 1.

paying the full cost[28], but they differ from the prevailing definition in *D. discoideum*. Cheating has been defined as the production of more than the fair share of spores, where fair share was defined as the mixing ratio between the participating strains[7]. This definition has been expanded and refined, but it was applied mostly to facultative cheaters, which cheat in the presence of a victim but form many spores during clonal development where no victim is present[29,30]. The *tgrB1⁻* cells produce few spores in pure populations and they do not produce more than 50% of the spores when mixed at equal proportions with the wild type. We therefore consider them to be obligatory partial cheaters, like the *fbxA⁻* obligatory cheater that cannot sporulate in a pure population but sporulates well and cheats in the presence of a victim[6]. Indeed, previous studies have already shown that *tgrB1⁻* cells occupy the prespore region in mixes with the wild type, but they were not considered cheaters because of the stricter definition[12]. We also note that *tgrC1⁻* cells produce very few spores when mixed with the wild type[18,20]. This observation is consistent with the distinct but complementary roles of *tgrB1* and *tgrC1*. As the greenbeard signal, *tgrC1* is only expected to contribute to the allorecognition aspects of the system but not to affect altruism or cheating directly.

*tgrB1* and *tgrC1* are both essential for development and cell-type differentiation, but they have different roles and the respective null-mutants have distinct phenotypes[12,15]. One of the relevant differences is the null-mutants' behavior when they are mixed with the wild type. The *tgrB1⁻* strain can cheat on the wild type because it is included in the mixed aggregates, whereas the *tgrC1⁻* mutant segregates from the wild type before tight aggregates are formed during development[12]. The inclusion of *tgrB1⁻* cells in the chimeric aggregate probably also explains how it can express the *cotB* and *ecmA* markers even though the greenbeard receptor function is lost. While *tgrB1* and *tgrC1* are essential for cell-type differentiation, they are not sufficient, and their signals can be bypassed by suppressor mutations[21,31,32]. In addition, *D. discoideum* cells employ numerous signals, both soluble and membrane-bound, to facilitate differentiation and morphogenesis[33]. We propose that *tgrB1⁻* cells, which are included in the chimeric aggregate, benefit from the signals produced by the wild type and can therefore produce spores and cheat. Our results therefore suggest that the greenbeard receptor function encoded by *tgrB1* is required for development in a non-cell-autonomous way.

Although *tgrB1* is highly polymorphic in natural populations[12], the mutations described here have not been identified in the sequenced natural strains[11,14]. Mutations that increase altruism are likely to be eliminated from the population during evolution because they would increase the fitness of their counterparts in mixed populations. Mutations that inactivate *tgrB1* cause cheating in mixed populations, but they probably get eliminated during evolution whenever the mutant cells develop clonally, due to the low sporulation efficiency of the mutant. We propose that the wild-type *tgrB1* alleles confer conditional altruism, which depends on reciprocal interactions between cells with matching *tgrB1-tgrC1* allotypes. This is, indeed, the property described as a greenbeard[1,4].

The availability of isogenic strains that differ only in their *tgrB1-tgrC1* allotype provided an opportunity to test the specificity of the greenbeard effect. The finding that the *tgrC1* signal identity determined the relationship between the cheater and its mixing partner supports the greenbeard hypothesis by showing that social interactions are restricted by the allotype. The reciprocal testing with the AX4 and QS31 allotypes showed that the effects are not specific to one allotype. Moreover, the absence of cheating by the strain that lacks both *tgrB1* and *tgrC1* showed that a compatible signal is required for cheating. These findings also show that this greenbeard system is based on the inclusion of kin rather than the exclusion of non-kin, reaffirming previous conclusions on *tgrB1-tgrC1* allotype recognition[13,17]. Separating the functions of the *tgrC1* signal from the *tgrB1* receptor revealed the role of each gene in the greenbeard locus. This separation was not possible in studies of *csaA*, the other *D. discoideum* greenbeard gene[10]. It also illustrates that a complex greenbeard locus can and does exist, despite early criticism that greenbeards were contrived and too complex to exist. Therefore, this study contributes significant empirical evidence to the growing body of support for the greenbeard hypothesis[4], including the seminal discovery of the *gp-9* greenbeard locus in fire ants[34].

Quantitative and mechanistic descriptions of greenbeard systems have been made in various biological systems, but many showed only partial evidence of the required properties – a rare perceptible signal, a specific receptor, and an altruistic action toward other organisms that exhibit the same signal[3,4]. The *tgrB1-tgrC1* system provides definitive empirical evidence for the existence of all the greenbeard properties in one locus.

## Methods
### Vectors
Cell-type specific markers: to generate the prestalk marker vector pDGB_A1N_ecmAp:mNeonGreen:mhcAt, we assembled the *ecmA* promoter, mNeonGreen coding sequence, and *mhcA* terminator as a transcriptional unit into the pDGB_A1N backbone using the GoldenBraid cloning method[35]. To generate a prespore marker vector we first cloned the *cotB* promoter by PCR amplification of a 1,718 bp fragment directly upstream of the *cotB* coding sequence from AX4 genomic DNA using the following primers: Forward: 5' gcgccgtctc actcgggagA CATTGTGTTA TTATTTGTGT GAAAAA 3' and Reverse: 5' gcgccgtctc actcgcattT TTATTACTGG TACTTTTACT ATATTAATGG TATATGTATA TGAGAT 3'. The first 19 bases of the 5' ends of each primer contain GoldenBraid-specific sequence grammar (lowercase), while the remaining bases are specific to the endogenous promoter sequence. This *cotB* promoter amplicon was domesticated into the pUPD2 backbone as described[35]. The vector pDGB_A1N_cotBp:mCherry:act8t was generated by assembling the *cotB* promoter, mCherry coding sequence, and *actin8* terminator as a transcriptional unit into the pDGB_A1N backbone using GoldenBraid[35]. We also generated a single hygromycin-based vector that carries a prestalk YFP marker and a prespore RFP marker. First, we generated another prestalk marker vector by assembling the *ecmA* promoter, eYFP coding sequence, and

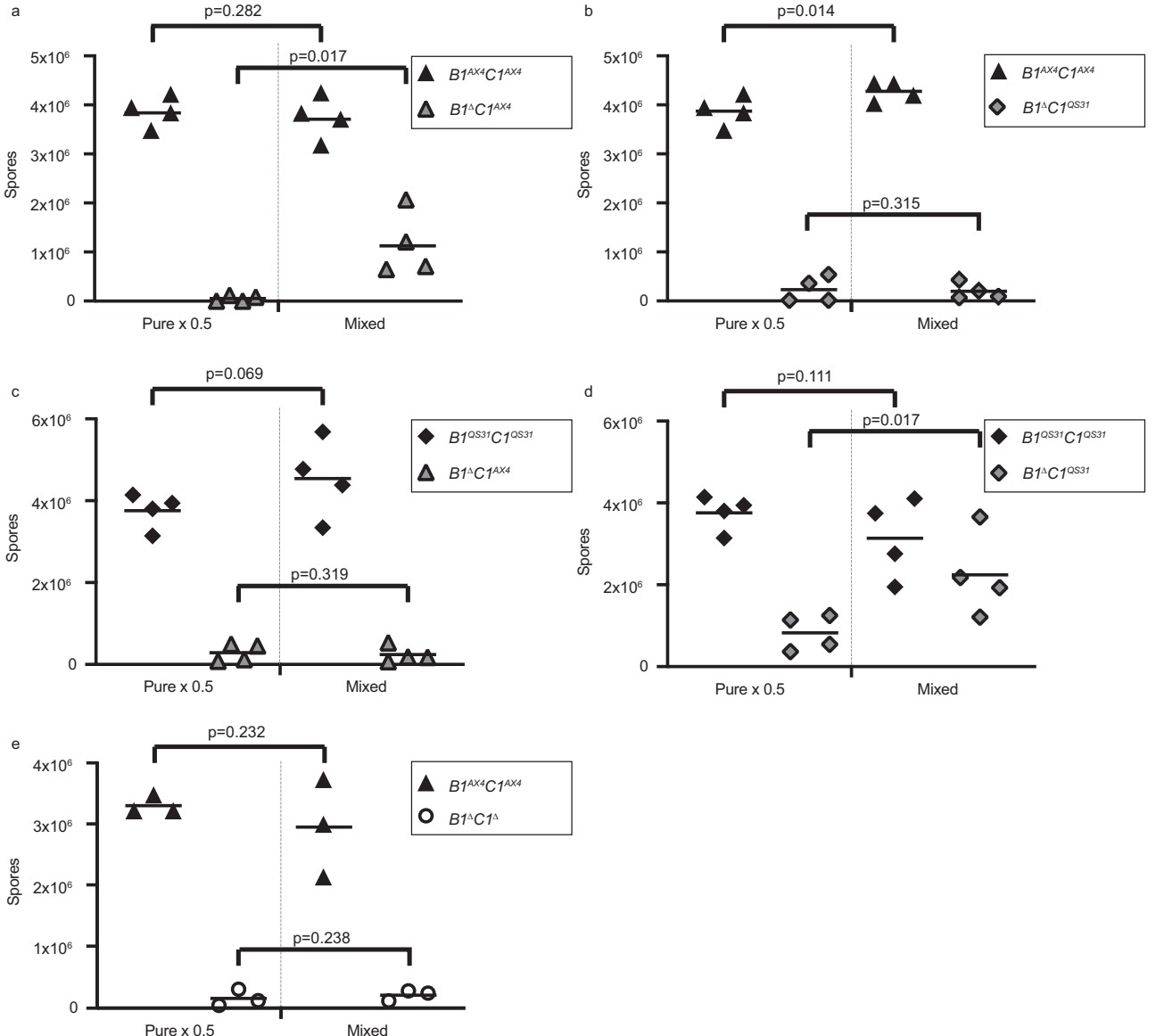

**Fig. 4 | *tgrB1⁻* cheating is allotype-specific.** We used strains that express constitutive GFP or RFP markers in which we deleted the resident *tgrB1-tgrC1* locus (*B1ᐃC1ᐃ*) and replaced it with a control locus from AX4 (*B1ᴬˣ⁴C1ᴬˣ⁴*) or a different allotype locus (*B1ᑫˢ³¹C1ᑫˢ³¹*). We also used the respective *tgrB1*-deletion strains (*B1ᐃC1ᴬˣ⁴* and *B1ᐃC1ᑫˢ³¹*). In each experiment we grew the strains separately, developed 7×10⁶ cells either in pure populations or mixed at equal proportions as indicated, and counted spores. The spore counts are shown as three or four independent replicates (symbols) and their averages (horizontal lines). The pure population counts were multiplied by 0.5 to scale them with the mixed population. Brackets and p-values (T-test, one sided, *n* = 3 in **e** and n = 4 in all the other panels)

compare the spore counts of each strain in the two conditions. **a** Matching *tgrC1* alleles from AX4 in a mix of wild-type *B1ᴬˣ⁴C1ᴬˣ⁴* and *tgrB1⁻* mutant *B1ᐃC1ᴬˣ⁴*. **b** Non-matching *tgrC1* alleles in a mix of wild-type *B1ᴬˣ⁴C1ᴬˣ⁴* and *tgrB1⁻* mutant *B1ᐃC1ᑫˢ³¹*. **c** Non-matching *tgrC1* alleles in a mix of wild-type *B1ᑫˢ³¹C1ᑫˢ³¹* and *tgrB1⁻* mutant *B1ᐃC1ᴬˣ⁴*. **d** Matching *tgrC1ᑫˢ³¹* alleles in a mix of wild-type *B1ᑫˢ³¹C1ᑫˢ³¹* and *tgrB1⁻* mutant *B1ᐃC1ᑫˢ³¹*. **e** A control mix of wild-type *B1ᴬˣ⁴C1ᴬˣ⁴* and the *tgrB1⁻ tgrC1⁻* double mutant *B1ᐃC1ᐃ*. The data for pure *B1ᴬˣ⁴C1ᴬˣ⁴* are identical between panels **a** and **b** and the data for pure and *B1ᑫˢ³¹C1ᑫˢ³¹* are identical between panels **c** and **d**. Source data are provided as a Source Data file.

*mhcA* terminator into the pDGB_A2N backbone. Then, we assembled it along with the pDGB_A1N_cotBp:mCherry:act8t vector into the pDGB_O1H backbone to generate pDGB_O1H_cotBp:mCherry:act8t; ecmAp:eYFP:mhcAt using GoldenBraid[35].

To generate a CRISPR/Cas9 vector for mutating the *tgrB1* gene, we used CRISPOR[36] to design an sgRNA targeting exon 1 of *tgrB1* immediately upstream of a PAM sequence found at base position 750. We annealed the following oligonucleotides and cloned into the pDGB_OH_CRISPR1 vector as described[35]. Sense: 5' agaagacgga gcaC-CAAAGC TCGATAAAAT GGA gtttcc gtcttct 3'; antisense: 5' agaagacgga aacTCCATTT TATCGAGCTT TGGtgctccg tcttct 3'. The 13 bases at either end of the oligonucleotide (lowercase) contain GoldenBraid-

specific grammar, while the intervening 20 bases (uppercase) constitute the guide sequence.

We also used the published vectors pDXA-tdTomato[13], ptgrB1:TgrB1AX4(L846F) bsR[15], ptgrC1:TgrC1AX4 bsR[15] and ptgrC1:TgrC1QS31 bsR[15]. Other fluorescent protein expression vectors, pDXA-mCherry, pDM304-mCherry, and pDXA-mCerulean, in which the fluorescent marker gene was driven by the *actin15* promoter, were a kind gift from Shigenori Hirose.

## Strains and strain construction

All the *D. discoideum* strains were generated by transformation of AX4[37] or its derivatives as detailed in Supplementary Table 1.

## Cell growth and transformation

We maintained *D. discoideum* cells at 22 °C in HL5 medium in a submerged culture and grew them for transformation and development in shaking suspension at 200 RPM with the adequate supplements and antibiotics[12]. We transformed the cells by electroporation, cloned by plating in association with bacteria and identified the desired clones by PCR analysis[13]. Before each experiment, we grew the cells at the logarithmic phase without antibiotics for 24 hours. Mutagenesis by CRISPR/Cas9 was performed as described[35].

We validated all the transformed strains by PCR and sequencing of the relevant genes. The PCR primers and sequencing oligonucleotides used for each diagnostic amplification, and sequencing oligonucleotides for CRISPR validation are listed in Supplementary Table 2.

## Development, imaging, and mixing experiments analysis

We induced development by washing the cells twice in KK2 (20 mM potassium phosphate, pH6.4) followed by starvation in a humid chamber at 22 °C. In mixing experiments, we grew the strains separately, washed the cells separately, counted them, and mixed in equal proportions before depositing them on solid substrates for development. To image developmental structures, we plated cells at a density of $2-5 \times 10^5$ cells/cm$^2$ on 1.5% Nobel agar made in KK2. Fluorescence and Differential Interference Contrast (DIC) microscopy images were captured with a Nikon (Tokyo, Japan) Eclipse Ti microscope using the NIS Elements 4.51.00 software[35]. The images shown are representative of at least 2 independent replications, each consisting of several hundred structures. To measure sporulation and cheating, we developed the cells on black nitrocellulose filters for 40 hours and harvested spores as described[38] with the following exceptions: we deposited $1.4 \times 10^6$ cells/cm$^2$ cells on each quarter filters and placed three replicate quarter filters on one filter pad per sample. After spore collection, we counted the spores and captured fluorescence and DIC micrographs of several fields to calculate the sporulation efficiency of each strain in the mix. The average of the three-quarter filter counts was reported as one data point. Each experiment was repeated at least three independent times. To compare between development in pure population and development in 1:1 mixes, we multiplied the pure population spore counts by 0.5 to scale them with the mixed populations. We performed one-sided paired T-tests using Microsoft Excel version 16.82 to compare pairs of pure and mixed populations.

## Reporting summary

Further information on research design is available in the Nature Portfolio Reporting Summary linked to this article.

## Data availability

All data generated or analyzed during this study are included in this published article and its supplementary information files. Source data are provided as a Source Data file. Source data are provided with this paper.

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

## Acknowledgements

We thank Shigenori Hirose for unpublished vectors, Ana Mesquita for carrying out preliminary experiments, and Elizabeth Ostrowski for discussions.

## Author contributions

M.K. and G.S. designed the experiments. M.K. performed the experiments. P.L. and M.K. generated vectors and strains. P.L. validated key experiments. G.S. wrote the manuscript. M.K. and P.L. edited the manuscript.

## Competing interests

The authors declare no competing interests.
