## [Peer Review File · Nature Communications]

The greenbeard gene *tgrB1* regulates altruism and cheating in *Dictyostelium discoideum*REVIEWER COMMENTS

Reviewer #1 (Remarks to the Author):

The manuscript by Katoh-Kurasawa et al examines the role of tgrB1 as a greenbeard gene in Dictyostelium. By using mutant and wild allele variants of the polymorphic tgrB1 gene and strain mixing experiments during starvation induced aggregative multicellular development, they make a case for its role in altruistic and cheating behaviors. The TgrB1-TgrC1 receptor-ligand pair provide a fascinating and relatively well-understood example of how microbes use self or allo-recognition to identify compatible sibling cells to unify together in tissues required for multicellular development. Elegant genetic and microscopy approaches were used and the manuscript is generally well written. Below are specific comments and suggestions.

Major concerns:

The overall model has internal inconsistencies that are not addressed. If TgrC1 is the ligand and TgrB1 is the receptor, then how can a tgrB1 deletion be rescued by mixing with a WT tgrC1-B1 strain? In other words, how can the tgrB1 deletion mutant restore development gene expression of the ecmA and cotB reporters when there is no receptor to perceive and transduce the TgrC1 signal (Fig. 3)? Perhaps a model could be added? Also, would a more congruent model propose that TgrB1 and TgrC1 each function as both a ligand and receptor? In so doing there is a mechanistic basis for how the tgrB1 deletion can be rescued for gene expression in during mixing experiments. Here, in the deletion mutant the 'TgrC1 receptor' binds to the TgrB1 ligand on the donor which in turn intracellular signals ecmA and cotB expression. If this model is flawed, can the authors provide another model/explanation?

To argue for altruistic behaviors dominant gain-of-function mutations in tgrB1, which no longer require the TgrC1 ligand for activation, were used. Here, in mixing experiments, a modest increase in spore formation is found for the WT, while no significant difference in mutant sporulation efficiency is found compared to monoculture development (Fig. 1 and S1). To argue for altruism it seems necessary that the tgrB1 mutants would need to lose fitness in the mixing experiment, but this is not found, so it's unclear how this is an altruistic act. Additionally, are tgrB1 L846F or G275D GOF alleles found in wild populations? If not, then any argument about altruism is artificial.

What happens when a tgr1 deletion strain is mixed with tgr1-L846F GOF strain? Presumably this mixture would show more robust altruistic/cheating phenotypes than shown in Figures 1 and 2.

Figure 3 provides evidence for extracellular complementation of reporter gene expression in a tgrB1 deletion mutant. In this regard are the exposure times/camera settings the same in panels A and B (should be stated in legend)? Additionally, can a developmental monoculture of the tgrB1 deletion strain with the two reporters be included as a control. This would help in the interpretation of the figure and it recognized RNASeq published data is already provided in Fig. S2.

What happens to ecmA and cotB gene expression in Trg1-L846F mutant? Presumably these genes are expressed during development.

Can the authors provide additional mechanistic insights into how the tgrB1 activation alleles work? For example, are the substitutions in conserved or polymorphic regions? Are they located in predicted cytoplasmic or extracellular regions?

Minor points:

Lines 72-73. The wording is confusing. To clarify after "dominant mutations that activate the tgrB1 gene" insert "product" or refer to TgrB1 as a protein. Similarly, as written: "These mutations were all in conserved nucleotides," suggest these are cis regulatory mutations that presumably effect tgrB1 expression, which is not accurate.

In Fig. 1 micrographs, it would assist the reader to label the "prestalk" and "prespore" regions in the developing slug.

Fig. 1e, 2d and 4 graphs on the y-axis they start at "0". If this is correct, then add breaks in axes to indicate six-log log jumps to next tick mark. Similarly, on line 130, the numeric meaning of "very few spores" based of Fig. 2d is unclear because of the poor resolution of the y-axis.

Lines 212-216 are not clear. Please expand the meaning of these explanations.

Line 218: This phrase is overstated "tgrB1- cells did not contribute to the prestalk and stalk tissues in mixes." Restate.

Reviewer #2 (Remarks to the Author):

Greenbeard genes are those that simultaneously encode three traits: (i) a signal; (ii) an ability to recognise the signal in others; (iii) altruism contingent on signal recognition. Previous work had shown that *tgrB1-tgrC1* genes in *D. discoideum* satisfy greenbeard properties i (signal production) and ii (signal recognition), but evidence for iii was indirect. This work builds on previous findings to show that *tgrB1-tgrC1* genes also exhibit greenbeard property iii (altruism contingent on signal recognition). Overall, I really like the paper – the experiments are convincing, and the discovery that *tgrB1-tgrC1* is a greenbeard has potentially general implications for our understanding of how / why altruism evolves. However, I have a few comments.

Firstly, the framing in the Abstract / Intro / Discussion seems to be a bit contradictory. You make statements like: "empirical studies have shown the existence of various greenbeard types in the real world" (L32); "Two greenbeard examples have been described in *D. discoideum*" (L37); "The developmental cell adhesion gene *csaA* is a greenbeard..." (L44); etc. However, I think it would be more accurate to say that these previous studies have identified "putative" (or "candidate") greenbeards, since they tend to lack conclusive evidence for each of the three required greenbeard traits (i-iii listed above). I think framing previous studies as having identified "putative greenbeards" rather than "greenbeards" is more accurate, and highlights the significance / novelty of the present study.

Secondly, you say that "[the study] illustrates that a complex greenbeard locus can and does exist, despite early criticism that greenbeards were contrived and too complex to exist" (L256). I agree with this, but it may give the impression that this is the first time that all three required greenbeard traits (i-iii) have been demonstrated for a single gene. I don't think this is the case – for instance, the *Gp-9* gene in fire ants has been shown to simultaneously encode a signal, a capacity to recognise the signal, and a trait that causes the killing of queens that lack the gene. I think a mention of the evidence for greenbeards outside of *D. discoideum*, and in particular the *Gp-9* gene in fire ants, would be useful to set the present study in a broader context.

Thirdly, you state on L237 that "Altruism genes, such as *tgrB1*, confer an apparent selective disadvantage that must be mitigated to stabilize cooperation". This is not true – altruism genes are easily favoured by kin selection when cells interact with highly related individuals such as clones, as is the case in *D. discoideum*. Your discussion of the possibility of pleiotropy stabilising cooperation in this system therefore seems unnecessary / superfluous, since there is no reason to think that cooperation should be disfavoured in the first place.

Finally, I have a series of minor suggestions for improving accessibility for a general audience / explaining jargon:

L59: "One study showed that allorecognition can protect cooperators against cheaters". Is it worth clarifying that this study examined cooperation encoded by another locus (i.e., not the *tgr*), to distinguish it from the present study?

L72: "dominant mutations that activate the *tgrB1* gene". Could you be more specific about what activate means here? Presumably, activation means that, in the cells where the dominant mutations are present, the *tgrB1* gains an ability to bind to any cell (and trigger downstream effects) irrespective of *tgrC1* identity?

L79: "wild-type counterpart, GFP-marked AX4". Is it worth clarifying that the wildtype strain has a pair of tgrB1-tgrC1 alleles (i.e., these are not knocked out / absent)? I wasn't sure at first.

L109: "To test that possibility, we co-developed differentially-labeled AX4 and tgrB1- cells." Is it worth highlighting in this section that the strains being mixed in this experiment have the same tgrC1 allele (which is why binding is possible)?

L227 - 230: I don't understand what "facultative cheaters" and "obligatory partial cheaters" means. In what sense is the cheating facultative as opposed to constitutive, or partial as opposed to complete? I think this needs more explanation, if the terminology is to be retained.

- Related to the above points, could it be worth adding a conceptual figure to show what tgrB1 and tgrC1 genes the two strains have, and whether tgrB1 is activated, in each experimental setting?

RESPONSE TO REVIEWERS' COMMENTS

We appreciate the positive comments made by the reviewers and thank them for their insightful and constructive criticism. We think that the revised manuscript is greatly improved thanks to these changes.

Below, we have included their comments verbatim followed by our respective responses. All the text changes are also tracked in the combined manuscript file. Changes made to the figures are described in the specific responses to the comments. All the line numbers refer to the final text version (using 'No Markup' as the tracking option in Microsoft Word). We also reformatted the manuscript according to the journal formatting instructions – some of those changes are not tracked. We note that the old Fig. S2 is now Fig. S3, because we added a new Fig. S2 (see below). Finally, we revised Figure 4 to correct a typographical error (replacing the p value on the bottom of Fig. 4b from 0.314 to 0.315). The rest of the original figures were not changed.

Reviewer #1 (Remarks to the Author):

The manuscript by Katoh-Kurasawa et al examines the role of tgrB1 as a greenbeard gene in Dictyostelium. By using mutant and wild allele variants of the polymorphic tgrB1 gene and strain mixing experiments during starvation induced aggregative multicellular development, they make a case for its role in altruistic and cheating behaviors. The TgrB1-TgrC1 receptor-ligand pair provide a fascinating and relatively well-understood example of how microbes use self or allo-recognition to identify compatible sibling cells to unify together in tissues required for multicellular development. Elegant genetic and microscopy approaches were used and the manuscript is generally well written. Below are specific comments and suggestions.

Major concerns:

The overall model has internal inconsistencies that are not addressed. If TgrC1 is the ligand and TgrB1 is the receptor, then how can a tgrB1 deletion be rescued by mixing with a WT tgrC1-B1 strain? In other words, how can the tgrB1 deletion mutant restore development gene expression of the ecmA and cotB reporters when there is no receptor to perceive and transduce the TgrC1 signal (Fig. 3)? Perhaps a model could be added? Also, would a more congruent model propose that TgrB1 and TgrC1 each function as both a ligand and receptor? In so doing there is a mechanistic basis for how the tgrB1 deletion can be rescued for gene expression in during mixing experiments. Here, in the deletion mutant the 'TgrC1 receptor' binds to the TgrB1 ligand on the donor which in turn intracellular signals ecmA and cotB expression. If this model is flawed, can the authors provide another model/explanation?

Author reply: We have considered that possibility in the past but found no evidence that TgrB1 and TgrC1 are co-receptors/co-ligands, so we do not favor this model. While we have no congruent model (yet) that explains all the properties of these proteins in development and in social behavior, there seems to be a good explanation for the

issues brought up by Reviewer #1. While *tgrB1* and *tgrC1* are necessary for development and cell-type specific gene expression, they are not the only signals that matter. Actually, there is good evidence (Li 2015, Li 2016, and Wang 2015) that the *tgrB1*-*C1* signals can be bypassed, so their integration might be non-cell autonomous. In addition, *D. discoideum* cells use many other signals to coordinate their differentiation and development (e.g., PSF, cAMP, CMF, SDF, DIF, STIF). Therefore, the absence of the TgrB1 receptor does not necessarily mean that the null-cells should be unable to differentiate. In fact, *tgrB1*-null cells can form rare tight aggregates very late in development (we added that fact in Fig. S4). Furthermore, in mixing experiments, *tgrB1*-null cells remain intermixed with the wild type during development, which exposes them to signals that might allow development and cheating (Fig. 2 in this manuscript, and Benabentos et al. 2009). On the other hand, *tgrC1*-null cells segregate from the wild type before tight aggregates are formed, so they probably cannot benefit from signals that are produced afterwards. We have added a paragraph to the Discussion section, explaining this issue and providing relevant references (lines 279-294).

To argue for altruistic behaviors dominant gain-of-function mutations in *tgrB1*, which no longer require the TgrC1 ligand for activation, were used. Here, in mixing experiments, a modest increase in spore formation is found for the WT, while no significant difference in mutant sporulation efficiency is found compared to monoculture development (Fig. 1 and S1). To argue for altruism it seems necessary that the *tgrB1* mutants would need to lose fitness in the mixing experiment, but this is not found, so it's unclear how this is an altruistic act.

Author reply: Even though the mutant cells do not exhibit significantly reduced sporulation efficiency, they do increase the sporulation efficiency of the wild-type partner by 24% in the case of L846F and 33% in the case of G275D. In the following generations, the mixing ratios between wild type and mutant would not be 1:1, as they were in the beginning, but $(1+a)^n:1$, where 'n' is the number of generations and 'a' is the increase rate (0.24 in the case of L846F and 0.33 in the case of G275D). Over 'n' generations of development in chimeras, these ratios will increase exponentially (frequency-dependance notwithstanding). Since the relevant niche must be limited, the wild type is expected to overwhelm the mutant within a few generations. Helping the wild type to the point of self-extinction or near extinction is undoubtedly altruistic. We have revised the Discussion section extensively to explain this critical point in a focused and explicit manner (lines 244-255).

Additionally, are *tgrB1* L846F or G275D GOF alleles found in wild populations? If not, then any argument about altruism is artificial.

Author reply: We have surveyed the sequences of 50 wild strains and did not find mutations in any of the positions that matched the various activated *tgrB1* alleles found in that screen, including the two mentioned in this manuscript (Li, 2016). We revised the entire paragraph (lines 63-84) and added the sentence: "None of the activating mutations found in the screen matched any naturally-occurring SNPs in the *tgrB1* coding sequences." (lines 77-78) to emphasize this point. While this is an important

point, we respectfully disagree with the reviewer that the 'argument about altruism is artificial'. The mutations we have generated/discovered in the screen are indeed artificial by definition, because they were made in the laboratory using a chemical mutagen, but the argument about the consequent altruism is not artificial. Laboratory experiments commonly use mutated alleles to investigate gene function, even though most of these mutations are not abundant in natural populations. The fact that the specific tgrB1 gain-of-function (GOF) mutations have not been found in wild populations is therefore irrelevant to the argument we are making. In fact, it might indicate that these mutations are short-lived in nature because they confer constitutive altruism and are therefore eliminated by natural selection after a few generations of mixing with wild-type cells. This argument is not essential for the manuscript, so we did not include it in the revised version.

What happens when a tgr1 deletion strain is mixed with tgr1-L846F GOF strain? Presumably this mixture would show more robust altruistic/cheating phenotypes than shown in Figures 1 and 2.

Author reply: We performed the proposed experiments and found that the mixture shows a more robust altruistic phenotype, as predicted by the reviewer, but the cheating was almost unchanged. We included the results in a new supplement figure (Fig. S2, lines 598-609). We also added text to describe the experiment (lines 151-158) and its interpretation (lines 163-164).

Figure 3 provides evidence for extracellular complementation of reporter gene expression in a tigrB1 deletion mutant. In this regard are the exposure times/camera settings the same in panels A and B (should be stated in legend)?

Author reply: We added the camera settings for all the images in the manuscript in Supplement File S1 and indicated that at the end in all the relevant figure legends. We note that the results shown in this case are qualitative, not quantitative, and our description in the text (lines 167-184) does not imply otherwise. These data are intended to show whether the reporter genes were expressed or not, and the position of the positive cells.

Additionally, can a developmental monoculture of the tgrB1 deletion strain with the two reporters be included as a control. This would help in the interpretation of the figure and it recognized RNASeq published data is already provided in Fig. S2.

Author reply: We performed the proposed experiments and included them in a new supplement figure (Fig. S4, lines 623-635). In that figure, we show a typical pure population of a wild-type structure at the finger stage, and a typical pure population of the tgrB1-minus strain, which forms loose aggregates at the same time. Most of the mutant cells do not express the marker genes, but we included arrows to point a few faint cells that do express the markers. We also included an image of a rare finger-like structure that is formed occasionally by the tgrB1-minus strain in late developmental times. All the images in the new figure were acquired at the same settings as the

images shown in Figure 3, so comparison of intensities is possible, even though we are not trying to make a quantitative argument. The camera settings are included in the new Supplement File S1. We also added a new paragraph (lines 185-193) to describe these findings in the relevant context.

What happens to *ecmA* and *cotB* gene expression in *Trg1-L846F* mutant? Presumably these genes are expressed during development.

Author reply: We agree with the reviewer that this is an interesting subject, but we do not think the results are relevant to the conclusions of this manuscript. We are in the process of measuring gene expression in this mutant using RNA-seq analysis. Preliminary data show that both *ecmA* and *cotB* are expressed in the mutant at the same level as the wild type but somewhat precociously. We plan to publish these results in the context of a larger ongoing transcriptome study of the *tgrB1-tgrC1* pathway.

Can the authors provide additional mechanistic insights into how the *tgrB1* activation alleles work? For example, are the substitutions in conserved or polymorphic regions? Are they located in predicted cytoplasmic or extracellular regions?

Author reply: We added the requested information in the introduction (lines 72-84). The L846F mutation is in the highly conserved cytoplasmic domain, next to a phosphorylation site of unknown significance, and the G275D mutation is in the first (most N-terminal) conserved immunoglobulin fold domain of the extracellular region. We did not discuss the mechanistic implications of these positions because *TgrB1* is the only protein in its class that has ever been studied, so we do not know the relationship between structure and function in this protein yet.

Minor points:

Lines 72-73. The wording is confusing. To clarify after “dominant mutations that activate the *tgrB1* gene” insert “product” or refer to *TgrB1* as a protein. Similarly, as written: “These mutations were all in conserved nucleotides,” suggest these are cis regulatory mutations that presumably effect *tgrB1* expression, which is not accurate.

Author reply: We added the word ‘product’ after *tgrB1* gene (line 69). We also explained that *tgrB1* is a polymorphic gene but the variability is not distributed evenly across the coding region. The mutations were all in nucleotides that encode invariable amino acids (none of the mutations were in cis-regulatory regions). The revised text in lines 72-84 explains that we are referring to the coding sequences of the gene and that the mutations were all in the invariable regions of the coding sequences, not overlapping with any known naturally-occurring SNPS.

In Fig. 1 micrographs, it would assist the reader to label the “prestalk” and “prespore” regions in the developing slug.

Author reply: The slug image in Fig.1a was labeled as anterior and posterior and the figure legend was modified to indicate that the anterior is mainly prestalk and the posterior is mainly prespore (lines 515-517).

Fig. 1e, 2d and 4 graphs on the y-axis they start at “0”. If this is correct, then add breaks in axes to indicate six-log log jumps to next tick mark. Similarly, on line 130, the numeric meaning of “very few spores” based of Fig. 2d is unclear because of the poor resolution of the y-axis.

Author reply: The y-axes in these graphs are linear and contiguous, starting at zero and increasing in regular increments as indicated, so it would be incorrect to add breaks or anything else. We revised the text (lines 148-150) to provide the numerical values of the respective averages, so there is no ambiguity. The final figures in the publication should provide a higher resolution that would allow the reader to discern the individual data point values from the graph.

Lines 212-216 are not clear. Please expand the meaning of these explanations.

Author reply: The discussion was aimed at explaining why the altruistic strain did not exhibit reduced sporulation despite its increased contribution to the stalk. We speculated that cells which normally fail to differentiate could join the spore-producing population to make up for the cells that altruistically became stalk cells. We gave two examples of cells that do not normally differentiate (‘nulls’ and ‘loners’), and provided the appropriate references. We also referred the reader to other possibilities that are not related to cell-type proportioning, which is a common term that describes the establishment and maintenance of prespore/prestalk ratios. We left these sentences as they were, now in lines 251-253 of the revised text, but we revised the context extensively (lines 244-255). We think that the major changes we made to the entire paragraph in response to the second comment made by Reviewer #1 should help clarify these sentences.

Line 218: This phrase is overstated “tgrB1– cells did not contribute to the prestalk and stalk tissues in mixes.” Restate.

Author reply: We toned the statement down by adding the word ‘significantly’ after the word ‘contribute’ (line 257).

Reviewer #2 (Remarks to the Author):

Greenbeard genes are those that simultaneously encode three traits: (i) a signal; (ii) an ability to recognise the signal in others; (iii) altruism contingent on signal recognition. Previous work had shown that tgrB1-tgrC1 genes in *D.discoideum* satisfy greenbeard properties i (signal production) and ii (signal recognition), but evidence for iii was indirect. This work builds on previous findings to show that tgrB1-tgrC1 genes also exhibit greenbeard property iii (altruism contingent on signal recognition). Overall, I really like the

paper – the experiments are convincing, and the discovery that *tgrB1-tgrC1* is a greenbeard has potentially general implications for our understanding of how / why altruism evolves. However, I have a few comments.

Firstly, the framing in the Abstract / Intro / Discussion seems to be a bit contradictory. You make statements like: “empirical studies have shown the existence of various greenbeard types in the real world” (L32); “Two greenbeard examples have been described in *D. discoideum*” (L37); “The developmental cell adhesion gene *csaA* is a greenbeard...” (L44); etc. However, I think it would be more accurate to say that these previous studies have identified “putative” (or “candidate”) greenbeards, since they tend to lack conclusive evidence for each of the three required greenbeard traits (i-iii listed above). I think framing previous studies as having identified “putative greenbeards” rather than “greenbeards” is more accurate, and highlights the significance / novelty of the present study.

Author reply: We added ‘putative’ before ‘greenbeard’ in the abstract (line 12) and main text (lines 29 and 33). In line 41 we replaced ‘*csaA* is a greenbeard’ with ‘*csaA* was described as a greenbeard’ because we already said it was ‘putative’ in the line 33.

Secondly, you say that “[the study] illustrates that a complex greenbeard locus can and does exist, despite early criticism that greenbeards were contrived and too complex to exist” (L256). I agree with this, but it may give the impression that this is the first time that all three required greenbeard traits (i-iii) have been demonstrated for a single gene. I don’t think this is the case – for instance, the *Gp-9* gene in fire ants has been shown to simultaneously encode a signal, a capacity to recognise the signal, and a trait that causes the killing of queens that lack the gene. I think a mention of the evidence for greenbeards outside of *D. discoideum*, and in particular the *Gp-9* gene in fire ants, would be useful to set the present study in a broader context.

Author reply: We revised the text as suggested and added a general reference to a comprehensive review of greenbeards and a specific mention and reference to the *gp-9* locus in fire ants (lines 308-311).

Thirdly, you state on L237 that “Altruism genes, such as *tgrB1*, confer an apparent selective disadvantage that must be mitigated to stabilize cooperation”. This is not true – altruism genes are easily favoured by kin selection when cells interact with highly related individuals such as clones, as is the case in *D. discoideum*. Your discussion of the possibility of pleiotropy stabilising cooperation in this system therefore seems unnecessary / superfluous, since there is no reason to think that cooperation should be disfavoured in the first place.

Author reply: We deleted the entire paragraph.

Finally, I have a series of minor suggestions for improving accessibility for a general audience / explaining jargon:

L59: “One study showed that allorecognition can protect cooperators against cheaters”. Is it worth clarifying that this study examined cooperation encoded by another locus (i.e., not the *tgr*), to distinguish it from the present study?

Author reply: We revised the text as suggested (line 56-57)

L72: “dominant mutations that activate the *tgrB1* gene”. Could you be more specific about what activate means here? Presumably, activation means that, in the cells where the dominant mutations are present, the *tgrB1* gains an ability to bind to any cell (and trigger downstream effects) irrespective of *tgrC1* identity?

Author reply: The activated TgrB1 does not need to bind a ligand in order to trigger the downstream effects. It is active without the ligand. We revised the text to explain that the activation mutations allow the TgrB1 protein to exert its receptor activity in the absence of a ligand (line 71-72).

L79: “wild-type counterpart, GFP-marked AX4”. Is it worth clarifying that the wildtype strain has a pair of *tgrB1-tgrC1* alleles (i.e., these are not knocked out / absent)? I wasn’t sure at first.

Author reply: We revised the text as suggested (line 95-96)

L109: “To test that possibility, we co-developed differentially-labeled AX4 and *tgrB1*- cells.” Is it worth highlighting in this section that the strains being mixed in this experiment have the same *tgrC1* allele (which is why binding is possible)?

Author reply: We revised the text as suggested (line 127-128)

L227 - 230: I don’t understand what “facultative cheaters” and “obligatory partial cheaters” means. In what sense is the cheating facultative as opposed to constitutive, or partial as opposed to complete? I think this needs more explanation, if the terminology is to be retained.

Author reply: The terms ‘facultative’ and ‘obligatory’ refer to the sporulation efficiency of the cheaters when there is no victim to cheat on. Obligatory cheaters do not sporulate (well) in the absence of a victim whereas facultative cheaters can sporulate rather well even in a pure population. We revised the text accordingly and added a familiar example of the pioneering *fbxA*-null facultative cheater (lines 269-272).

- Related to the above points, could it be worth adding a conceptual figure to show what *tgrB1* and *tgrC1* genes the two strains have, and whether *tgrB1* is activated, in each experimental setting?

Author reply: We have used cartoon images of cells and molecules in previous publications to illustrate mechanistic aspects of the tgrB1-C1 signaling pathway. In this case, we prefer not to complicate the figures because the molecular mechanism aspects have already been published and they are not the main thrust of the manuscript.

REVIEWERS' COMMENTS

Reviewer #1 (Remarks to the Author):

The authors provided a thorough response to reviewer comments, which included a substantial amount of new data. Overall this is a nice manuscript of broad interest and is much improved. Below are two suggestions for further improvement and clarity.

The scales on the y-axis for viable spore counts need to be corrected on some figures. They start at "0" and then the first tick mark is 1,000,000 or similar. For example, in Fig. 4A, the B1(delta)/C1AX4 mutant (shaded triangles) shows that in pure culture this strain does not sporulate, i.e. all triangles reside at 0. Other figures have a similar problem. However, the text indicates these strains actually sporulate at relatively high numbers, which contradicts the data shown in some figures.

The manuscript would be enriched with an added paragraph in the Discussion about the various phenotypic alleles of *tgrB1* and their social consequences in nature. For example, although null alleles of *tgrB1* can lead to a cheating phenotype they are less fit in both mixed or in monocultures as compared to wild type (Fig. 2), thus they would be selected against in natural populations. As commented, the GOF alleles, which exhibit the altruistic phenotype, would also be selected against because they lose fitness in multicellular development. It should also be noted that the known wild-type alleles are 'proto-altruistic' genes because they do not behave in an altruistic manner.

Reviewer #2 (Remarks to the Author):

The authors have responded satisfactorily to each of my comments. It is a nice paper and I have no further comments.

RESPONSE TO REVIEWERS' COMMENTS

Reviewer #1 (Remarks to the Author):

The scales on the y-axis for viable spore counts need to be corrected on some figures. They start at "0" and then the first tick mark is 1,000,000 or similar. For example, in Fig. 4A, the B1(delta)/C1AX4 mutant (shaded triangles) shows that in pure culture this strain does not sporulate, i.e. all triangles reside at 0. Other figures have a similar problem. However, the text indicates these strains actually sporulate at relatively high numbers, which contradicts the data shown in some figures.

Author's Response: As we have already explained in our response to the original review, the scales of the y-axes in all the figures are correct, as are the individual data points and the averages. These y-axes are linear and they are indeed divided into increments of 1,000,000 or 2,000,000 (as dictated by the results). The sporulation data shown close to the zero value are indeed very low because they represent very low sporulation efficiencies. To address this issue even more explicitly, we have now added an Excel spreadsheet with all the source data, so there is no question about the data accuracy. We also added a sentence in the text, next to the first description of the low sporulation levels of the tgrB1-null strain. In that sentence we included a comment that the sporulation efficiency of this strain is less than 3% (this is because we plated 7,000,000 cells and recovered fewer than 200,000 spores). While a yield of 200,000 spores is a big number (compared to zero), it is actually rather low considering the size of the input. The most relevant conclusions of these figures is the increase in sporulation in the mixed population relative to the pure population. We therefore added the comment in the first instance, in the text that refers to Fig. 2d, but did not elaborate on this matter throughout the rest of the manuscript.

The manuscript would be enriched with an added paragraph in the Discussion about the various phenotypic alleles of tgrB1 and their social consequences in nature. For example, although null alleles of tgrB1 can lead to a cheating phenotype they are less fit in both mixed or in monocultures as compared to wild type (Fig. 2), thus they would be selected against in natural populations. As commented, the GOF alleles, which exhibit the altruistic phenotype, would also be selected against because they lose fitness in multicellular development. It should also be noted that that the known wild-type alleles are 'proto-altruistic' genes because they do not behave in an altruistic manner.

Author's Response: We agree with the reviewer. We added the following paragraph to the discussion (third to last paragraph): "Although tgrB1 is highly polymorphic in natural populations (12), the mutations described here have not been identified in the sequenced natural strains (11,14). Mutations that increase altruism are likely to be eliminated from the population during evolution because they would increase the fitness of their counterparts in mixed populations. Mutations that inactivate tgrB1 cause cheating in mixed populations, but they probably get eliminated during evolution whenever the mutant cells develop clonally, due to the low sporulation efficiency of the mutant. We propose that the wild-type tgrB1 alleles confer conditional altruism, which

depends on reciprocal interactions between cells with matching tgrB1-tgrC1 allotypes. This is, indeed, the property described as a greenbeard (1,4).